# Data-Driven Damage Quantification of Floating Offshore Wind Turbine Platforms Based on Multi-Scale Encoder–Decoder with Self-Attention Mechanism

Musa Bashir [1], Zifei Xu [1], Jin Wang [1] and C. Guedes Soares [2,*]

[1] School of Engineering, Faculty of Engineering and Technology, Liverpool John Moores University, Liverpool L3 3AF, UK
[2] Centre for Marine Technology and Ocean Engineering (CENTEC), Instituto Superior Técnico, Universidade de Lisboa, 1649-004 Lisboa, Portugal
[*] Correspondence: c.guedes.soares@centec.tecnico.ulisboa.pt

**Abstract:** A Multi-Scale Convolutional Neural Network with Self Attention-based Auto Encoder–Decoder (MSCSA-AED), is a novel high-performance framework, presented here for the quantification of damage on a multibody floating offshore wind turbine (FOWT) structure. The model is equipped with similarity measurement to enhance its capability to accurately quantify damage effects from different scales of coded features using raw platform responses and without human intervention. Case studies using different damage magnitudes on tendons of a 10 MW multibody FOWT were used to examine the accuracy and reliability of the proposed model. The results showed that addition of Square Euclidean (SE) distance enhanced the MSCSA-AED model's capability to suitably estimate the damage in structures operating in complex environments using only raw responses. Comparison of the model's performance with other variants (DCN-AED and MSCNN-AED) used in the industry to extract the coded features from FOWT responses further demonstrated the superiority of MSCSA-AED in complex operating conditions, especially in low magnitude damage quantification, which is the hardest to quantify.

**Keywords:** data-driven technique; damage quantification; floating offshore wind turbine; FOWT predictive maintenance; Convolutional Neural Network; multi-scale information fusion

## 1. Introduction

Rapid development of Floating Offshore Wind Turbines (FOWTs) for clean power generation is an important step in the strategy to ease global energy crises and mitigate the impact of climate change [1]. Review of the trend of wind farms shows that there is a recent tendency for floating wind farms to be installed at farther distances from the coast, as opposed to the shallow water fixed wind turbines developed in the first round of development [2]. The current trajectory of FOWT development can be grouped into two categories, namely, geometry-dependent operations and maintenance-driven operations. Continuous growth in FOWTs' size has become a major consideration in selecting FOWT foundation types and location.

Increase in FOWT sizes brings other significant challenges to installation, operations, and maintenance (O&M) and the attendant consequence is the requirement for exponential structural strength to adequately withstand the loads imposed by the differences in geometrical scale and environment loads. This is in addition to the fundamental need for safe and damage resilient operation. Therefore, it is important to conduct failure analysis of the wind platforms [3,4] to assess the probabilities of failure of the various components. The reliability of the floating platforms is essential to ensure that appropriate methods are adopted for this purpose [5].

There are available databases of failure and maintenance data, but this has been developed mainly for onshore platforms and, eventually, for fixed offshore platforms [6,7].

Therefore, it is important to use the existing data and modify it to make it applicable to off-shore turbines, as proposed by Li and Guedes Soares [8]. It should be noted that availability of appropriate failure and maintenance data is essential to establishing robust maintenance plans. These plans can be the basic age-based preventive maintenance (Santos et al. 2015) for opportunistic strategies [9,10] or for strategies that account for the logistic time of spare parts and vessel availability [11,12]. Condition-based maintenance has become more widely adopted [13] and specific approaches have also been developed for floating offshore turbines [14]. The route to improving maintenance plans is by increasing the condition-based maintenance by using more extensive structural health monitoring of the platforms and turbines. The direct relationship between structural health of components and FOWT safety typically affects how the platform can be operated and maintained to reduce the levelized cost of electricity (LCoE) [15].

Lack of effective maintenance methodology for offshore wind turbines, especially for deep water applications, is generally acknowledged as a significant obstacle to the development of FOWT technology. This is mainly due to the lack of failure and maintenance data in which to base maintenance plans and to the difficulties arising from lack of access for inspection and repairs, leading to high O&M costs [16,17]. Consequently, having an effective monitoring strategy and a robust quantification method for defects in FOWT components can provide a credible path for developing an intelligent method for preventive maintenance, based on actionable condition and operational data [18]. This approach allows rational decision-based maintenance planning and scheduling to be formulated as part of the proactive maintenance strategy, which, ultimately, improves safe operation, and reduces the maintenance costs and the LCoE of FOWTs.

Recent progress on the internet of things (IoT) and artificial intelligence (AI) technologies further strengthen the case for possible adopting of preventative maintenance in FOWTs. Machine learning (ML), a subset of AI, has become a mainstream approach in data-driven methodologies for preventative maintenance in the industries. The data-driven technique for structural damage prediction, a quantitative method for detection and diagnosis of damages or faults, has recently been the subject of interest to researchers. Some notable studies on data-driven techniques have been reported, including the works of Cho et al. [19] who developed a detector to locate the structural damage on blades of a wind turbine by using Kalman filters and artificial neural networks.

Dao [20] established a regression model for wind turbine condition monitoring and the reliable detection of anomalies in their operations. Yang et al. [21] used independent component analysis and Mahalanobis distance to quantify damages in the transmission system of a wind turbine. Rai et al. [22] applied empirical mode decomposition (EMD) and k-medoids clustering to assess the structural health of wind turbine bearings, which is helpful in estimating the residual capability of the wind turbine bearings. Cheng et al. [23] used a Convolutional Neural Network (CNN) to estimate the degradation of the mechanical systems from raw signals using a complete ensemble empirical mode-based label. Guo et al. [24] adopted a CNN with linear labels to supervise the network and quantify damage degradation. In a further similar work, Guo et al. [25] used a Recurrent Neural Network (RNN) with kurtosis-based nonlinear labels to construct the degree of damages on mechanical parts of wind turbines. Chen et al. [26] studied different types of RNNs to extract damage features only using prior experience to realize damage quantification.

However, as summarized in these papers, although many ML (including DL) methods have been used to identify faults and defects, their algorithms largely rely on defects or faults labels. The process of labeling these defects or faults is very labor-intensive and requires engineers to manually annotate the data. From an efficiency perspective, the process is time-consuming and laborious. It is also dependent on prior knowledge when being looked at from the model's perspective.

Recent attempts by researchers to expand the state of knowledge in this subject include investigation on methodologies for damage quantification based on unsupervised learning. Peng et al. [27] combined particle filter and Deep Belief Network (DBN) to calculate

the damage magnitude on structures of wind turbines. Dai et al. [28] used Generative Adversarial Network (GAN) to achieve a structural monitoring of wind turbine structure without employing any manual supervision.

Guo et al. [29] used a multi-scale network with an attention mechanism to extract features and obtain information about damage magnitude. Suh et al. [30] designed a GAN, with a U-net architecture, to deal with the damage quantification of a rotating machine and to estimate its residual capability. To enhance the performance of convolutional networks and make them more powerful and accurate for decision-making in maintenance, Guo et al. [31] introduced a multi-scale feature extraction process in an auto encoder–decoder to achieve the development of unsupervised architecture. The method was successfully applied in a rotating machine mounted on a fixed platform. This study pointed out that the feature fusion of a decoder had a consequential effect on the performance of the Neural Network (NN) model for damage quantification. Although a supervised deep learning network (DLN) model can be quickly adapted to quantify the known damage magnitude, the labeling process is time-consuming and labor-intensive. This approach is inefficient and almost unsuitable for industrial-scale application because of the difficulty to supervise and label all data in practical engineering applications.

Notwithstanding, the unsupervised models are dedicated to learning and establishing the response patterns of the machines in the healthy state. It is still possible to quantify the damage magnitude by comparing the similarity between the baseline and test coded data in which the challenges of these methods are the robustness of an unsupervised model and the performance of the chosen similarity function. More importantly, for a system such as FOWT, the environment in which it operates further contributes to the complexity in platform dynamic responses, as opposed to a generic rotating system on a fixed platform. This presents a precisely unique challenge in how to successfully incorporate the unsupervised learning method in diagnosis and prognosis for a system operating in such complex and unstable environments. At present, there are few studies on intelligent damage estimation for FOWTs' towers and tendons.

Existing data-driven (quantitative) damage models are predominantly built using supervised learning. These models are not particularly suitable for floating offshore wind turbine damage quantifications for maintenance, due to rigorous demands for accurate feature labeling. Application of unsupervised learning in damage quantification and its robustness in floating wind turbines have not yet been investigated. Therefore, this study was inspired by the possibility of developing a remarkable algorithm that reduces human interface in damage quantification by using CNN with a self-attention mechanism. Consequently, this research developed a high-performance framework, named Multi-Scale Convolutional Neural Network with Self Attention-based Auto Encoder–Decoder (MSCSA-AED), to quantify the damage magnitude of the 10 MW FOWT. The proposed method enables a faster convergence with smaller errors than the baseline models currently used in the industry. The main features in the encoder and decoder having the same scale are reevaluated by the self-attention mechanism to improve efficiency and quality of features for damage quantification. The MSCSA-AED model is trained by only using the normal (healthy) data from raw platform responses, an important capability that minimizes prognostic latency. The MSCSA-AED model is optimized to enhance clarity in utilizing its parameters. Following the optimization, the quantified damage values of the 10 MW FOWT is calculated by the similarity function implemented in the model. The reliability of the damage quantification model using different similarity functions was examined in this study to support both practical and theoretical references for damage quantification of 10 MW FOWTs. Hence, the main contributions of this research are summarized below:

(1) This research developed an unsupervised approach, based on deep neural network with self-attention mechanism, named MSCSA-AED, to intelligently extract features (healthy and damaged) to establish the state of health of a 10 MW FOWT from the coded features. The method provided a path to adding the multiscale information present in typical multi-scale resolution features of the FOWT's responses (due to its

complex operating environment) for efficient damage quantification in maintenance planning and scheduling.

(2) The research found that the influence of applying different similarity functions to establishing the reliability of damage quantification of the FOWT was significant. The capability to analyze the characteristics of the platform's different degrees of freedom (DOFs) when implemented in the model improved accuracy and reliability in quantifying structural damage in complex operating environments.

(3) This study used a 10 MW FOWT with tendon damage simulated data as a case study to examine the performance of the proposed MSCSA-AED method on damage quantification. The reliability of the proposed method was demonstrated through examination of different damage scenarios and operating conditions with tendon defects. It was found that the SE metric-based quantification method offered more reliable results than other methods do.

Following an introduction in Section 1, the rest of the paper is organized as follows. Section 2 presents details of the FOWT structure, damage scenarios and simulations. Section 3 presents the proposed damage quantification methods. Section 4 discusses both quantitative and qualitative results as part of the validation process. Section 5 concludes the research.

## 2. The 10 MW FOWT Geometry, Damage Conditions and Simulations

### 2.1. Structural Configuration of the 10 MW Multibody FOWT

This study used a novel TELWIND concept of the multibody 10 MW FOWT developed by Esteyco [32] for the ARCWIND project [33]. The platform consists of a telescopic tower supported by an upper tank (UT) for buoyancy and a lower tank (LT) for ballasting. The UT and LT are connected by 12 tendons (steel cables), as shown in Figure 1. Three mooring cables connected to the fairleads at the top surface of the UT, located at 14 m below mean sea level, are used for station keeping of the platform. Structural ribs are added to the platform to provide strength and sufficient space to connect the upper and lower tanks. This further ensures that the 10 MW FOWT remains safe and stable in the event of a limited tendons rupture.

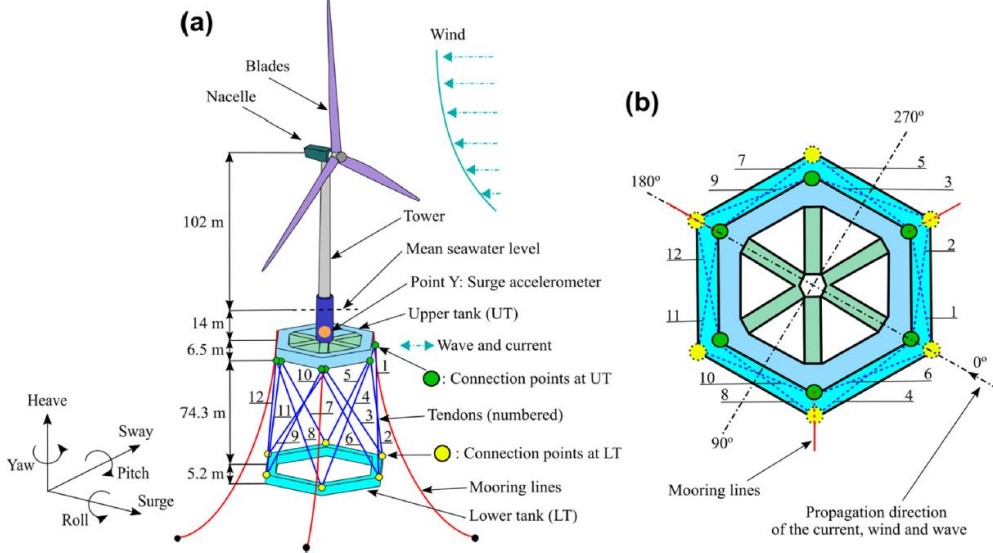

**Figure 1.** The 10 MW multibody FOWT. (**a**) Fully assembled floating wind turbine showing all components and position of load measurement. (**b**) The plan view of the multibody platform.

Details of the model properties include a draught of 20.5 m at the upper tank with a total mass of $5.31 \times 10^6$ kg and a displaced volume of 7399.02 m$^3$. The lower tank has a

draught of 100 m and the total mass is $8.72 \times 10^6$ kg, which equals a volume of 7922.92 m$^3$. The 12 tendons have a collective pretension that is approximately equal to $3 \times 10^6$ N.

### 2.2. The Damage Modeling on the 10 MW FOWT

Damage in the structural components of the FOWT model were examined under varying Environmental and Operating Conditions (EOCs), corresponding to seven conditions. This comprised 4 wind speeds (WS), namely, WSs 4 m/s, 11.4 m/s, 18 m/s and 25 m/s with 4 significant wave heights (Hs), namely, Hs 1.61 m, 2.16 m, 2.95 m and 4.02 m (Table 1).

**Table 1.** The varying EOC details.

| Wind Speed (m/s) | Significant Wave Hight (m) | Peak Frequency (Hz) | Propagation Direction of Sea, Current and Wind (°) | Current Speed (m/s) | Excitation Bandwidth (Hz) |
|---|---|---|---|---|---|
| 4 | 1.61 | 0.285 | 0 | 0.22 | [0.1–100] |
| 11.4 | 2.16 | 0.185 | 0 | 0.22 | [0.1–100] |
| 18 | 2.95 | 0.14 | 0 | 0.22 | [0.1–100] |
| 25 | 4.02 | 0.112 | 0 | 0.22 | [0.1–100] |

The FOWT begins generating power when it reaches its cut-in speed of 4 m/s. The rated condition of the 10 MW FOWT is 11.4 m/s of the operating speed at which it is expected to optimally generate power. The FOWT's cut-out speed is 25 m/s, above which the FOWT is shutdown to protect it from adverse environmental effects.

The wind condition used in this is study was generated using the Kaimal spectrum [34], based on different time series of wind excitation and corresponding to the varying wind speed regime effects on FOWT. Consequently, to ensure that there was consistency of results in the simulations, the same spectral intensity was maintained throughout the simulations. The irregular waves components of the varying EOCs were generated using the modified two-parameter Pierson–Moskowitz spectrum [35,36]. Furthermore, a current of constant speed and direction was included in the prediction of the EOC. Table 1 and Figure 1b present the details of the EOCs' wind, wave and current parameters used in this research. These parameters represented normal and the most severe EOC features at the selected site in the offshore area of northern Scotland.

For this investigation, the platform was simulated under healthy and damaged conditions to predict the corresponding responses. The healthy conditions were simulated under four kinds of wind speed conditions. The damaged conditions were modeled, based on stiffness (%) reduction in the tendon. The main benefit of this approach was that any amplified damages, that reflected the platform dynamics, could be measured with a degree of certainty. This was subsequently used to examine the effects of EOCs under given WS and Hs on the structural health condition of the FOWT. The stiffness reduction ranged from 10% to 100 % with an increment of 10%, in which 100% stiffness reduction represented a state of total failure of the tendon. Following the platform simulations, it was established that Tendon 6 (T6) and Tendon 8 (T8) represented the extreme cases, and they were subsequently selected for examination. Tendon T6, located in the wave direction, suffered the largest tension, while Tendon T8, located close to a mooring line, was randomly selected (Figure 1b). Although the fatigue damage on tendons was outside the scope of this study, some fatigue results were considered in the study to investigate the process of quantifying invisible damage or defects on the tendon. Thus, the datasets obtained using stiffness reduction of 10–90% for training and 10–60% for testing were used in this paper for damage quantification investigation.

### 2.3. Simulations of Platform Response

Prediction of raw vibration responses under varying EOCs was undertaken to obtain datasets corresponding to healthy and different damaged conditions of the 10 MW multi-body FOWT. The responses were obtained from a coupled numerical simulation [37] of the

platform (including upper and lower tanks), mooring lines and tendons as a flexible multi-body structure. Simulations were conducted using an in-house developed coupling tool, F2A, which combined the beneficial capabilities of ANSYS-AQWA and NREL FAST [38], to accurately predict the FOWT's dynamic responses.

An important consideration in the operations of the platform is its stability if a tendon fails. For this reason, the FOWT platform was designed to have a non-exceedance pitch motion response variation range of −15 deg to 15 deg under both healthy and damaged states (Figure 2). This was further confirmed by the inclusion of redundancy in tendon design in the event of a tendon or a fairlead failure. The goal was solely to protect the overall integrity of the platform and ensure that remaining tendons had sufficient reserve capacity to withstand catastrophic failure. Further details on other properties of the platform, such as natural frequencies and eigenmodes, are provided in reference [39,40]. To demonstrate the accuracy and reliability of the proposed MSCSA-AED, this study examined damages using datasets from the three most dominant (surge, heave, sway,) of the six degrees of freedom accelerations. The responses were measured at Point Y (Figure 1a). The frequency of the sampled acceleration signals was $f_s$ = 10 Hz (acceleration signal bandwidth of [0–5] Hz) with each having $N$ = 20,000 samples (2000 s). Similarly, an acceleration response dataset was selected from each of the simulations to correspond to the DOFs for evaluation of the healthy conditions. Details of the simulations and measured signals are presented in Table 2.

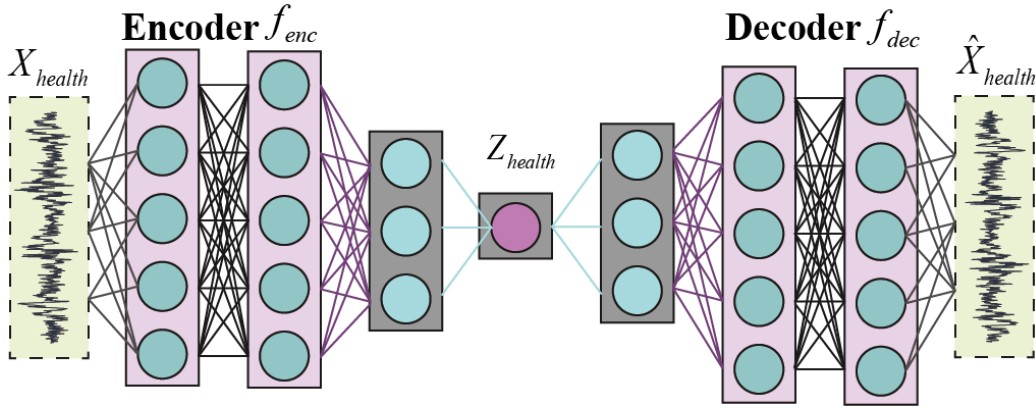

**Figure 2.** The DCN-AED architecture.

**Table 2.** Simulation cases and corresponding vibration signals.

| Structural Description State | Number of Damaged Tendons | Number of Damage Magnitudes | Number of Simulations |
|---|---|---|---|
| Healthy | - | - | 4 (one under each wind speed [4, 11.4, 18, 25] m/s |
| Damaged | (Tendons 6, 8) | 19 | 80, One per damage magnitude [10:10:100] % under each wind speed [4, 11.4, 18, 25] m/s on each tendon |

Sampling frequency: $f_s$ = 10 Hz, acceleration signal bandwidth: [0–5] Hz. Signal length: $N$ = 20,000 samples (2000 s).

Table 2 presents the healthy and damaged conditions. In this study, only the responses of the healthy condition were used in the training of the data-driven model. The damaged conditions datasets were used for the damage quantification and to examine the accuracy and reliability of the model.

## 3. Damage Quantification Method

### 3.1. Problem Description

In this study, an NN-based Auto Encoder–Decoder (AED) was designed as a communication tool for quantifying the damage magnitude in FOWT tendons. The tool uses the distance

calculated by similarity functions between the coded features extracted by NN-AED of the health condition to quantify the total damage magnitude. The mathematical representation uses $D_{health}$, which denotes a health condition within a domain $X_{health}$, and the raw responses (acceleration) for the operation. The training data ($X_{health} = \{x_{health}^i\}_{i=1}^N$) sampled from $X$ are sent into the encoder $f_{enc}$ to obtain the coded features $Z_{health} = f_{enc}(X_{health}|\theta_{enc})$, where $Z_{health} = \{z_{health}^i\}_{i=1}^N$, $\theta_{enc}$ is the parameter of the encoder, and $z_i = f_{enc}(x_{health}^i|\theta_{enc})$. The coded features are fed into the decoder, $f_{dec}$, to reconstruct the $\hat{X}_{health} = \{\hat{x}_{health}^i\}_{i=1}^N = f_{dec}(Z_{health}|\theta_{dec})$.

The main goal for adopting this approach in this study was to minimize the distance between the $X$ and $\hat{X}$, and obtain the optimal parameters of the model. Consequently, the mean square error (MSE) was used as a loss function to optimize the parameters of encoder $\theta_{enc}$ and the decoder $\theta_{dec}$ by gradient descent. The similarity between healthy and damage conditions changed when the damage magnitude increased. However, this change was unclear and remained subtle, due to the significant influence of the complex operating conditions of the FOWT. Thus, the damage in the tendon structure was quantified by using the distance between the coded features $Z_{health}$ from healthy conditions in the domain $X$, and the damage coded features $Z_{damage} = \{z_{damage}^i\}_{i=1}^N$, where $z_{i\ damage}^N = f_{enc}(x_{i\ damage}^N|\theta_{enc})$, $X_{damage} = \{x_{damage}^i\}_{i=1}^N$ were obtained from the unknown health condition in the domain $X_{damage}$. Therefore, damage quantification was defined as: $dist = dtw(Z_{health}, Z_{damage})$, where $dtw$ is a function used to calculate the difference between the two inputs, which can be either the root sum of squared differences, the sum of absolute differences or the square of the Euclidean metric. The $dist$ is the quantified value of the damage magnitude. The Euclidean Distance (ED) was used as the dtw function in the baseline model.

### 3.2. Baseline Model

The baseline model to extract the coded features was developed using a Deep Convolutional Neural Network (DCN)-based auto encoder–decoder. An illustration of the network is shown in Figure 2.

In the model training process, the coded features $Z_{health}$ were extracted by the encoder $f_{enc}$, which consisted of a series of convolutional kernels, batch normalization and ReLU activation from the input $X_{health}$. The coded features $Z_{health}$ were resized by the decoder $f_{dec}$ that consisted of sets of deconvolutions, batch normalization and leaky ReLU activation to rebuild the $\hat{X}_{health}$. The parameters of the DCN-AED model were optimized by a gradient descent optimizer, based on the Mean Square Error (MSE) function, as: $L(X_{health}, \hat{X}_{health}) = \frac{1}{N}\sum_{i=1}^N (X_{health} - \hat{X}_{health})^2$.

### 3.3. Proposed Model

A Multi-Scale Convolutional Neural Network with Self Attention based Auto Encode–Decoder (MSCSA-AED) was proposed to extract coded features from the tendon responses in this study. The MSCSA-AED model was equipped to fuse the multi-scale characteristics in the encoder and decoder with a view to improving the accuracy of the reconstructed signals. In addition, a multi-head Attention, one of the famous self-attention mechanisms, was used to compare the information in the encoder with those in the decoder and to generate the attention features for the decoder to rebuild a more reliable signal. The architecture of the MSCSA-AED is presented in Figure 3.

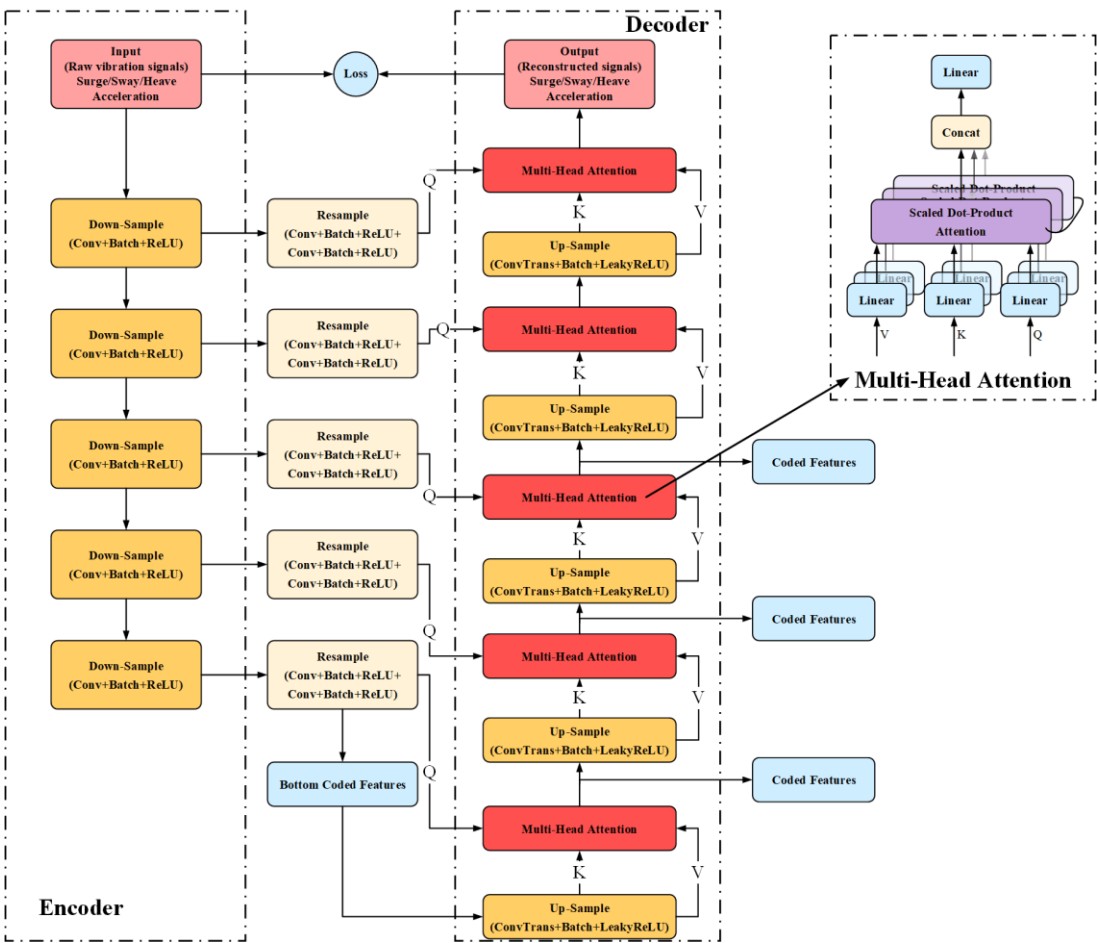

**Figure 3.** The MSCSA-AED architecture.

The MSCSA-AED model consists of a multi-scale resolution encoder–decoder and a multi-head attention mechanism. The input raw signals are down sampled by the encoder, while the neural network with different depths conducts the multi-scale resolution sampling process. The coded features of different scales are used to quantify the damage and are further investigated in subsequent studies. However, the coded features in the deepest layer are designated as default in comparison to other types of AEDs needed to examine the effectiveness and reliability of the proposed MSCSA-AED. The uniqueness of the proposed MSCSA-AED is that it is capable of considering the correlation between the encoded and decoded features by the multi-head attention mechanism. A comparison of the encoded features of resampling with the decoded features to obtain the weighted features for signal reconstruction is shown in the highlighted part of Figure 3. The process assumes that the feature corresponding to each round of the up-sampling is $Z_{up}^i$, and the resampling feature connected to the down-sampling is $Z_{down}^i$. By considering $Z_{up}^i$ as Key $K$, with Value $V$, and resampling $Z_{down}^i$ as the Query $Q$, the attention features can be obtained through the Multi-Head Attention mechanism, and the attention features of each layer are regarded as the coded features of different scales. The head attention is computed as:

$Attention(Q, K, V) = softmax\left(\frac{QK^T}{\sqrt{d}}\right)V$, where $d$ is the dimension of the $Q$, $K$ and $V$.

The procedure for training the MSCSA-AED model $G$ with similarity distance for damage quantification is summarized in Algorithm 1, in which the parameters of $G$ are defined.

---

**Algorithm 1** MSCSA-AED module with Similarity Function estimation for damage quantification

---

**1. Obtain parameters $\theta$ including weights and bias of the MSCSA-AED model $G$**

    **Input:** $X_{health}$ and initial parameters $\theta$

    **Output:** $\hat{\theta}$      - the estimation of the MSCSA-AED model
    **While $i \leq Epochs$** (*i.e., number of epochs*) **do**
        **Repeat**
            $\theta \leftarrow$ Update parameter in model $G$ via ADAM
            Descent
        **Until** *convergence of $\theta$*
    **End**
**End**
**2. Estimate the Similarity function between coded features of baseline $Z_{health}$ and current $Z_{health}$**
    **Input:** $X_{health}$, $X_{damage}$ and model $G$
    **Output:** *Damage Quantification*
    Estimation the coded features: $\hat{Z}_{health} = G(\hat{\theta}|X_{health})$ and $\hat{Z}_{damage} = G(\hat{\theta}|X_{damage})$
    Estimate the Similarity between coded features with different coded features $dist = dtw(Z_{health}, Z_{damage})$ to quantify the damage
**End**

---

### 3.4. Damage Quantification Methods

The main goal for developing the MSCSA-AED was to quantify the structural damage on tendons of the FOWT. The flowchart of the damage quantification based on the proposed MSCSA-AED model is shown in Figure 4, which presents the methodology for the proposed MSCSA-AED model with similarity function to quantify the damage without requiring any prior knowledge or human interface.

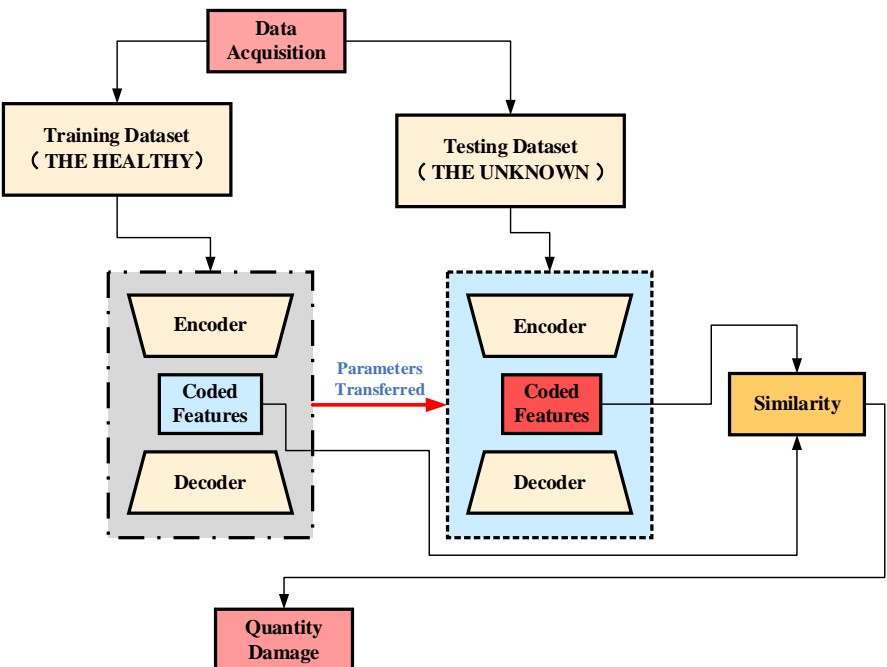

**Figure 4.** The MSCSA-AED based damage quantification framework.

The AED was trained using the dataset collected from the healthy FOWT without any structural damage under different EOCs. In the testing phase, the unknown data collected from the same sensor, which might have been from a damaged structure, was processed by the trained AED to obtain the unknown coded features. The damage quantification was addressed by comparing the similarity of the healthy coded and damaged features.

## 4. Discussion and Analysis

This section discusses and analyzes the reliability and validity of the proposed method in quantifying structural damage on wind turbines by using tendons as a case study. In the performance examination, the influence of hyper-parameters and wind on operating conditions, along with the corresponding coded features, were all investigated from the deepest part of the AED. The features from the deepest (bottom) part were selected because they are most useful in quantifying the damage since the noisy information would be removed by filters in the encoder. The healthy and damaged datasets were collected by the sensor on the tower base part of the upper tank.

### 4.1. Hyper-Parameters Examination

The hyper-parameters adopted for the examination of the proposed MSCSA-AED model were grouped in four cases, provided in Table 3 This included learning rate, mini-batch size and maximum epochs. All the optimized parameters were determined as those which had the best validation loss. In this examination, surge acceleration was used as the input features of the model.

**Table 3.** Hyper-parameters list for the proposed model.

| Hyper-Parameters | Learning Rate | Mini Batch | Max-Epoch |
|---|---|---|---|
| Case 1/Case 1$'$ | 0.001/0.005 | 50 | 200 |
| Case 2/Case 2$'$ | 0.001/0.005 | 100 | 250 |
| Case 3/Case 3$'$ | 0.001/0.005 | 200 | 350 |
| Case 4/Case 4$'$ | 0.001/0.005 | 300 | 500 |

Figure 5 presents the damage quantification using the MSCSA-AED model with a Euclidean Similarity function and trained under different hyper-parameters. The results show the presence of small fluctuations on damage quantification that varied with increase in damage magnitude when the stiffness reduction was larger than 30%. The results of the MSCSA-AED model trained using "Case 3 and Case 3$'$", showed more robust capability in estimating the damage magnitude, which appeared to be monotonic when using the method to quantify other damages corresponding to stiffness reduction of 30%, 50% and 70%. When the real damage magnitude was lower than 30%, the model with "Case 3 and Case 3$'$" hyper-parameters offered advantages in that quantification obviously increased in a monotonous way with increase in stiffness reduction. The outcome indicated a positive contribution from the MSCSA-AED model to quantifying the damage on the FOWT tendon. However, in Case3$'$, there was no significant change, with the real damage magnitude varying from 0.1 to 0.2. Thus, the hyper-parameters with learning rate 0.001, mini-batch 200 and maximum epoch 350 were set as the preferred parameters to train the model. Analysis of the model's complexity is presented in Table 4.

**Table 4.** Complexity analysis of the models.

| Method Name | Complexity Index |
|---|---|
| DCN-AED | O $(4.7 \times 10^5)$ |
| MSCNN-AED | O $(3.3 \times 10^7)$ |
| MSCSA-AED | O $(1.7 \times 10^7)$ |

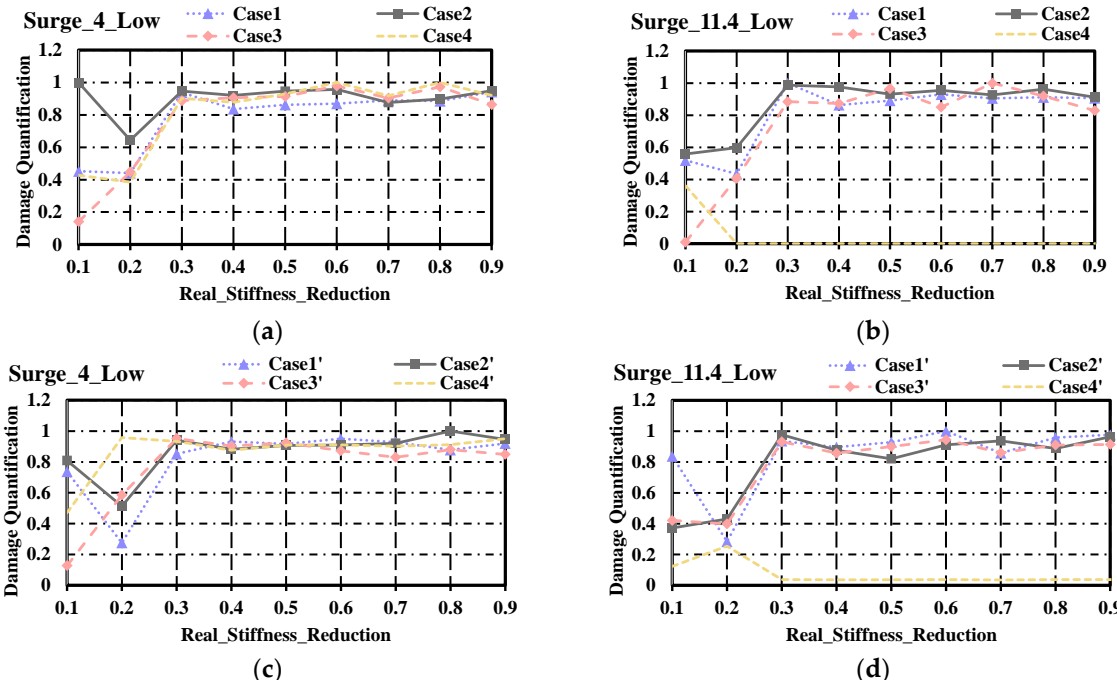

**Figure 5.** Damage quantification by MSCSA-AED with Euclidean similarity examined under different hyper-parameters. (**a**) Surge acceleration under 4 m/s, measured by the lower sensor using learning rate, was 0.001. (**b**) Surge acceleration under 11.4 m/s, measured by the lower sensor using learning rate, was 0.001. (**c**) Surge acceleration under 4 m/s, measured by the lower sensor using learning rate, was 0.005. (**d**) Surge acceleration under 11.4 m/s, measured by the lower sensor using learning rate, was 0.005.

The computational complexity of the proposed method was significantly higher than DCN, but lower than MSCN because of the introduction of recurrent neural networks in the feature resampling. The proposed method could obtain better performance at a relatively low complexity when it was combined with the performance analysis results of the quantified impairment to further demonstrate its superiority.

### 4.2. Damage Quantification Based on Different DOF Features

The six tendons, which had stiffness reduction from 10% to 90%, were quantified by several models, including DCN-AED, MSCNN-AED and the MSCSA-AED, to examine their reliability in damage quantification. All these models were trained using "Case 2" hyper-parameters and with the Euclidean distance used to measure the similarity between their respective coded features of healthy and structurally damaged conditions. The results of the examined cases for surge, sway and heave are presented in Figure 6.

As shown in Figure 6, by using surge acceleration as the feature to train a model, the proposed MSCSA-AED model showed the best capability in quantifying damages on the tendons of the FOWT. In Figure 6b, small fluctuations on damage quantification that varied with damage magnitude when the stiffness reduction was larger than 30%, were observed. The results showed that MSCSA-AED model could estimate the damage magnitude with a high degree of accuracy, as seen in damage conditions with monotonicity, and when quantifying damages that corresponded to stiffness reduction of 30%, 50% and 70%. Furthermore, a comparison of the performance results of the DCN-AED and MSCNN-AED models, when the real damage magnitude was lower than 30%, being the most challenging cases in real engineering application, was conducted. The results showed that the MSCSA-AED model offered superior capability in damage quantification for subtle damages (low stiffness reduction cases). In the case of using sway and heave accelerations features to train a model, the proposed MSCSA-AED model performed very

well, as demonstrated by its capability to quantify the damage of the tendons of the FOWT. Figure 6c–f present comparisons of the damage quantification conducted by the MSCSA-AED model with monotonous responses and those from the DCN-AED and MSCNN-AED models. This further showed that the proposed model was very capable of quantifying damages in low stiffness cases for different DOFs, as demonstrated by using the stiffness reduction method.

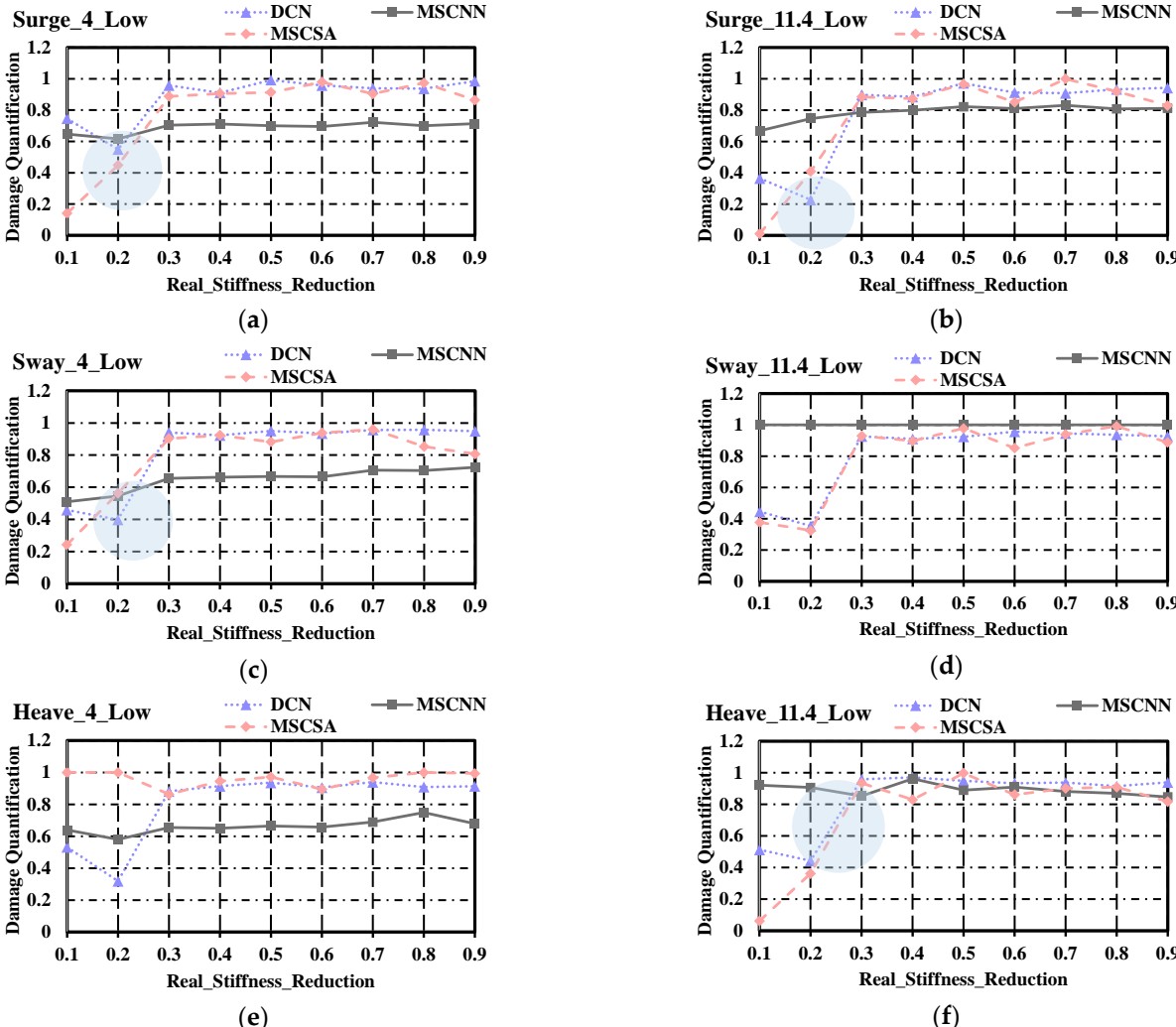

**Figure 6.** Damage quantification by Euclidean similarity. (**a**) Surge acceleration under 4 m/s, measured by the lower sensor. (**b**) Surge acceleration under 11.4 m/s, measured by the lower sensor. (**c**) Sway acceleration under 4 m/s, measured by the lower sensor. (**d**) Sway acceleration under 11.4 m/s, measured by the lower sensor. (**e**) Heave acceleration under 4 m/s, measured by the lower sensor. (**f**) Heave acceleration under 11.4 m/s, measured by the lower sensor.

Monotonicity was used as a metric to evaluate the performance of the quantified damage. As shown in Figure 7, the damage quantified by the proposed MSCSA-AED model offered the highest monotonicity score. Generally, the higher the damage monotonicity indicator, the more reliable and easier it was for the prognosis to be conducted. Hence, the comparative results indicated that MSCSA-AED (MSCSA) had a monotonicity of 0.6349. This was higher than the monotonicity of both DCN and MSCNN by at least 22.49%, indicating that the proposed method was well-suited to produce reliable damage quantification.

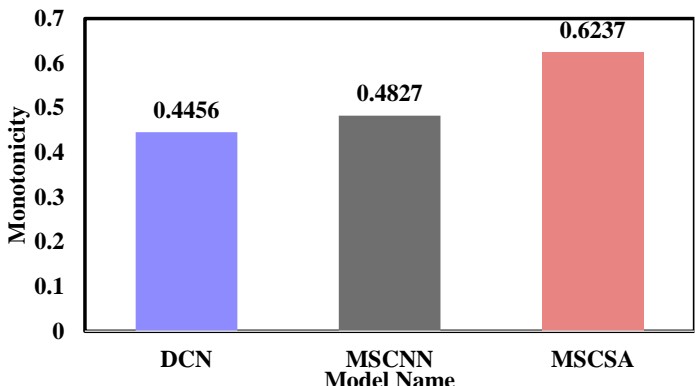

**Figure 7.** Monotonicity of the quantified damage using three models.

### 4.3. Quantification under Different Wind Conditions

In this section, structural damages on T6 and T8 were respectively quantified using the proposed MSCSA-AED model. Comparison between the DCN-AED and MSCNN-AED was conducted for the case studies investigated in this research to demonstrate the reliability of the model.

The results of these examinations from simulations conducted using the proposed MSCSA-AED with Euclidean distance to quantify the damages on the FOWT's tendons are presented in Figure 7. The effectiveness of the proposed method was examined under 4 m/s, 11.4 m/s 18 m/s and 25 m/s operating conditions, in which the 11.4 m/s was the FOWT rated condition, and the 25 m/s was the shutdown condition. By comparing plots of Figure 8a,b, based on surge acceleration, the results showed that damages to tendons T6 and T8 were correctly quantified under a smaller wind speed (4 m/s) effect than in the rated (11 m/s) condition in which the quantified damage changed more regularly with the actual damage. When the wind speed increased, as shown in Figure 8c, the quantified damage showed an increasing trend with a small damage condition (less than 30%), while irregular fluctuations occurred when the damage became large. This might have been caused by the surge response, which was easily affected by non-stationary winds. In this situation, other features, such as sway and heave, should be investigated to reach a conclusive decision. In Figure 8g–k, the proposed method accurately quantified the trend of damage for tendons T6 and T8, respectively. The results showed less instability and fluctuations when the actual damage exceeded 30%.

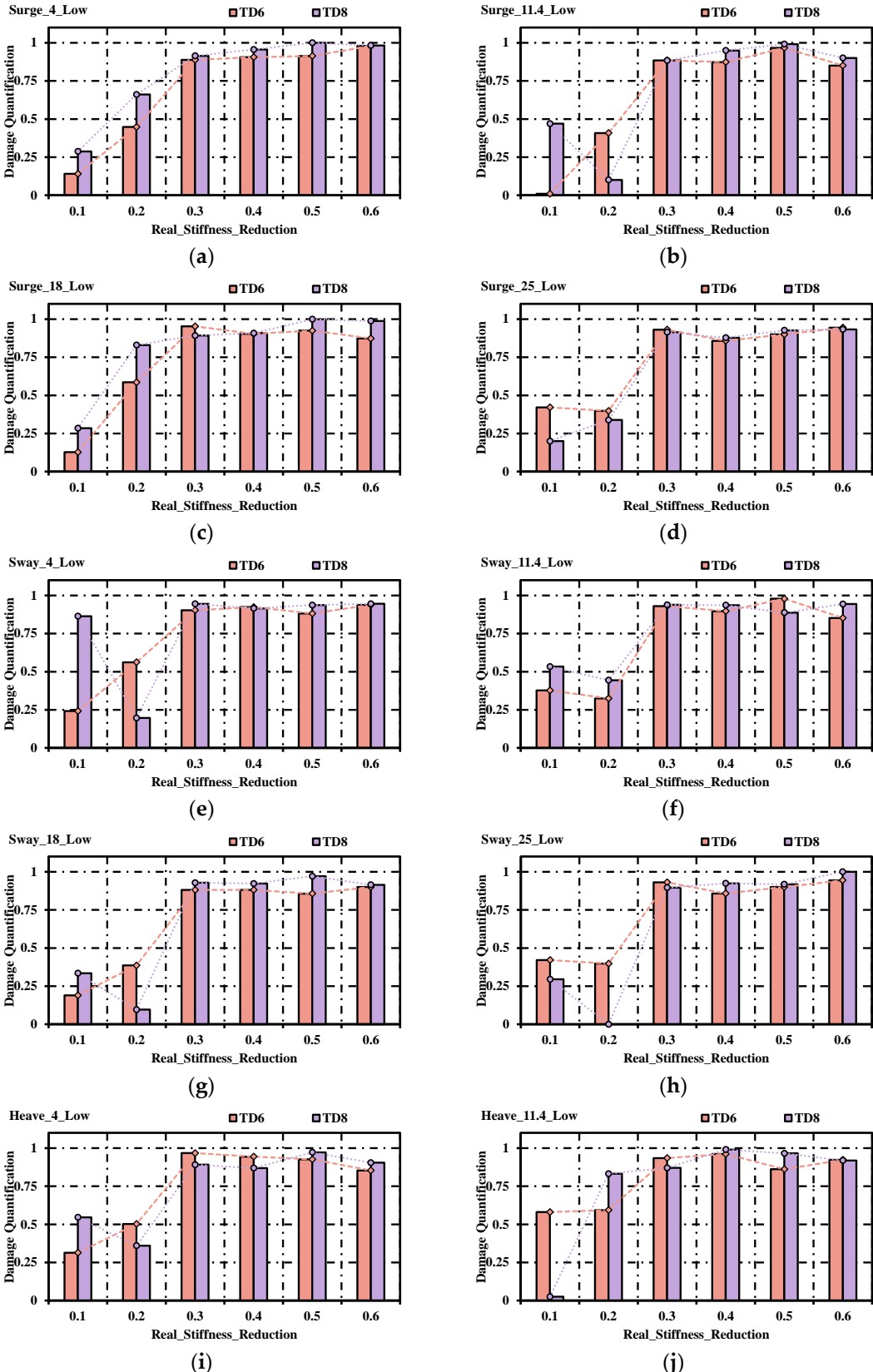

**Figure 8.** *Cont.*

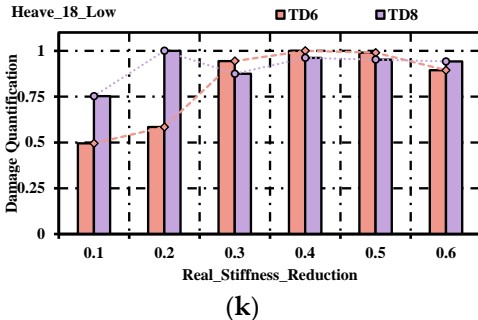
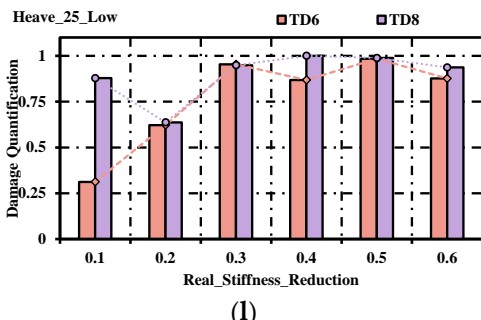

<div align="center">(<b>k</b>)           (<b>l</b>)</div>

**Figure 8.** Damage quantification by Euclidean similarity. (**a**) Surge acceleration under 4 m/s, measured by the lower sensor. (**b**) Surge acceleration under 11.4 m/s, measured by the lower sensor. (**c**) Surge acceleration under 18 m/s, measured by the lower sensor. (**d**) Surge acceleration under 25 m/s, measured by the lower sensor. (**e**) Sway acceleration under 4 m/s, measured by the lower sensor. (**f**) Sway acceleration under 11.4 m/s, measured by the lower sensor. (**g**) Sway acceleration under 18 m/s, measured by the lower sensor. (**h**) Sway acceleration under 25 m/s, measured by the lower sensor. (**i**) Heave acceleration under 4 m/s, measured by the lower sensor. (**j**) Heave acceleration under 11.4 m/s, measured by the lower sensor. (**k**) Heave acceleration under 18 m/s, measured by the lower sensor. (**l**) Heave acceleration under 25 m/s, measured by the lower sensor.

### 4.4. Quantification Using Different Similarity Measurements with Multi-Scale

The effectiveness of the damage quantification, based on the proposed different damage depth coded features in the MSCSA-AED with different similarity measurements, are examined in this section. Based on the surge acceleration, the Euclidean distance (E), Square Euclidean distance (SE) and Cosine (C) similarity were used to estimate the differences of the coded features.

In Figure 9, Depth_1 to Depth_4 were the coded features in the MSCSA-AED model with different multi-scale resolutions. The quantification of the coded features decreased with the deepening of the MSCSA-AED network. Hence, this confirmed the necessity to examine the reliability of the coded features with different scales (depths) when quantifying damage. Cosine similarity, Figure 8a–d, was used to estimate the distance between the coded features of healthy and damaged conditions to quantify the damage. When the wind speed was 4 m/s or 11.4 m/s, the coded features from deeper layers decreased at first and then increased in response to small structural damage quantification (less than 30%). The reason for this behavior was that the lowest level of coded features only included the down-sampling process. However, the effect of damage quantification became more accurate even when the wind speed was 25 m/s, and the result quantified using the deepest coded features had an opposite tendency but still with a consistent trend. In Figure 9e–h, Euclidean distance was used to quantify the damage. The trend of quantification results varied with real damage, and it was affected by the depth of coded features. Compared with C-based damage quantification, Elucidation-based damage quantification seemed more unstable. However, the effect of quantification depended not only on the depth (scale) of coded features, but also on the method of similarity calculation used. In Figure 9i–l, the Squared Euclidean (SE) distance was used to quantify the damage. It was found that, although the quantification results, corresponding to coded features of different depths (scales), occurred in opposite trends, the result was different to those from the deepest coded features in estimating damage which made the SE function more suitable to use than the C function. It can be seen that using SE to measure the distance of coded features to quantify the damage was more robust than the other methods.

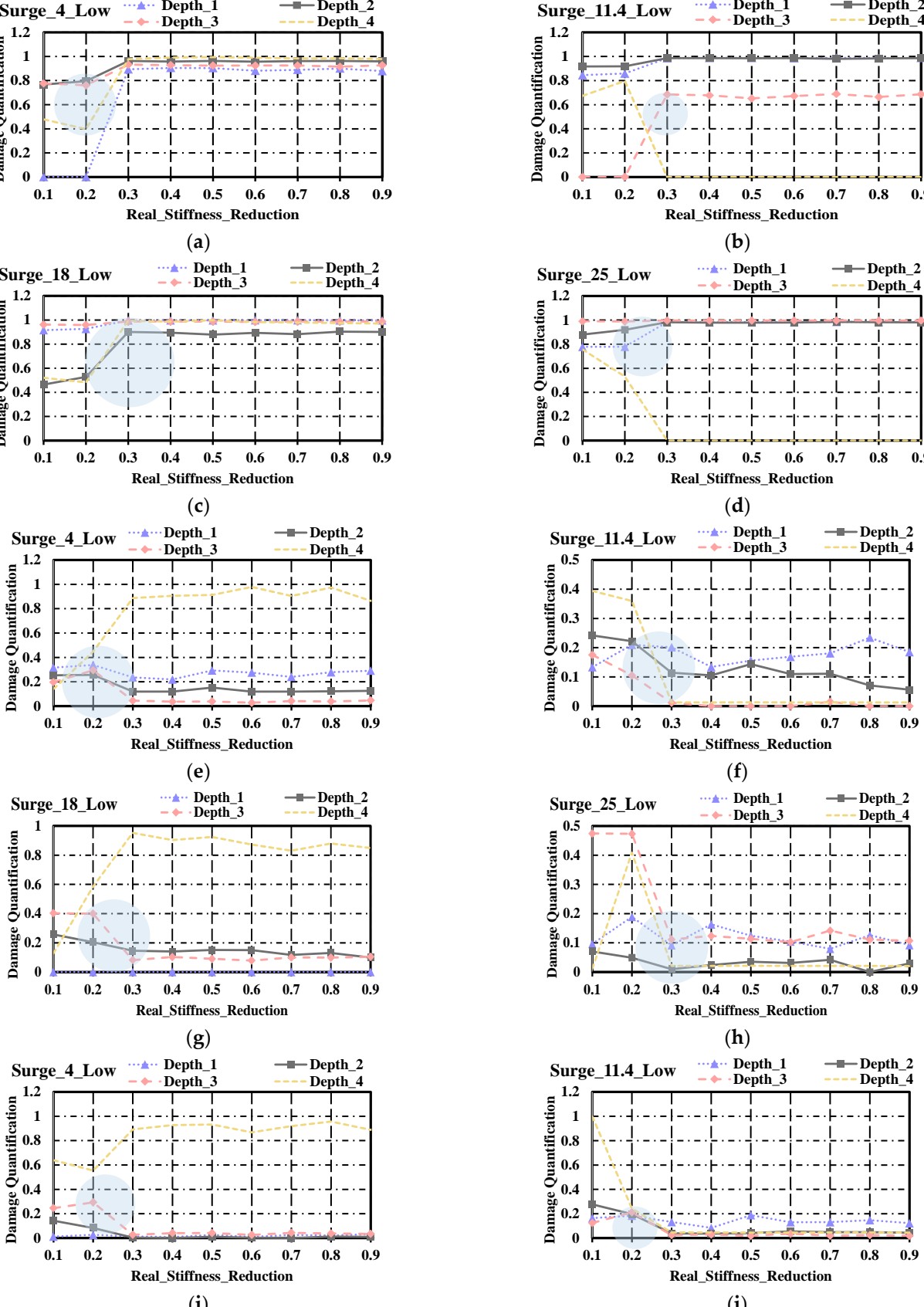

**Figure 9.** *Cont.*

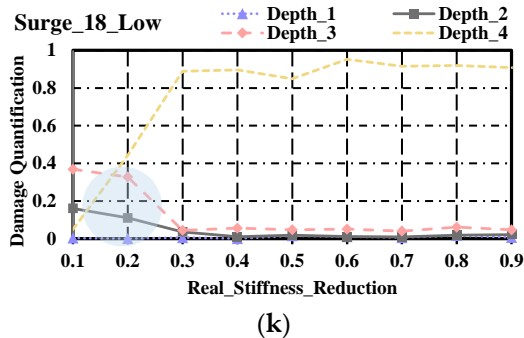
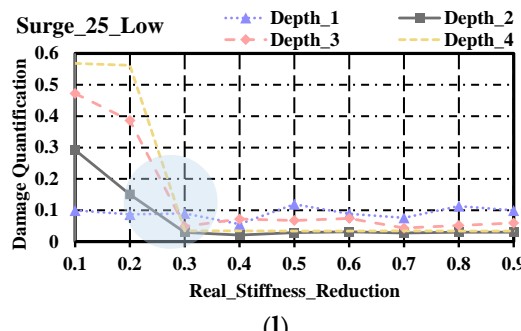

**Figure 9.** Comparison of different similarity functions. (**a**) Under 4 m/s, measured by C. (**b**) Under 11.4 m/s, measured by C. (**c**) Under 18 m/s, measured by C. (**d**) Under 25 m/s, measured by C. (**e**) Under 4 m/s, measured by E. (**f**) Under 11.4 m/s, measured by E. (**g**) Under 18 m/s, measured by E. (**h**) Under 25 m/s, measured by E. (**i**) Under 4 m/s, measured by SE. (**j**) Under 11.4 m/s, measured by SE. (**k**) Under 18 m/s, measured by SE. (**l**) Under 25 m/s, measured by SE.

## 5. Conclusions

Occurrence of damage on any floating wind turbine structural member has a significant impact on its safety and integrity. The methodology for estimation and quantification of damage can be used to assess the severity of the damage and its impact on the operations of FOWT. This is to ensure that safe and stable operation of the wind turbine is guaranteed. In this study, a novel data-driven method was developed for quantification of structural damages in FOWTs. The proposed MSCSA-AED model is used to obtain the coded features for use in diagnosis and quantification of damages. The model is coupled with similarity measurement to enhance its capability to accurately quantify the damage from raw operating responses and without human intervention. Case studies using different damage magnitude on tendons of a 10 MW multibody FOWT were conducted. Simulated datasets were used to examine the accuracy and reliability of the proposed method. The study found that addition of Euclidean distance enhanced the proposed MSCSA-AED model's capability to suitably estimate the damage in systems with complex response. Comparison of DCN-AED and MSCNN-AED models to extract the coded features from FOWT responses further showed that the proposed model offered more monotonous trend in low magnitude damage quantification, being the hardest to quantify, from all wind conditions. By studying the effects of coded features on different scales (depths) and the effects of different similarity measurement methods on damage quantification, it was found that using SE offered the best improvement in quantifying the damage. Furthermore, using the proposed model with SE produced more monotonic trend than other methods (Cosine and Elucidation) in the quantification of the damage. This is a significant benefit to accurate damage prediction, quantification, and evaluation of structural safety of floating wind turbines.

A limitation of the proposed method was that uncertainty analysis was not considered. Future research on uncertainty quantification and embedding of physics model in data-driven model for damage quantification is recommended.

**Author Contributions:** Conceptualization, Z.X.; methodology, Z.X.; software, J.W.; validation, J.W. and M.B.; formal analysis, M.B.; investigation, Z.X.; resources, C.G.S.; data curation, C.G.S.; writing—original draft preparation, Z.X.; writing—review and editing, M.B.; visualization, Z.X.; supervision, J.W. and C.G.S.; project administration, C.G.S.; funding acquisition, C.G.S. All authors have read and agreed to the published version of the manuscript.

**Funding:** This project was funded by the European Regional Development Fund (ERDF), Interreg Atlantic Area (grant number: EAPA_344/2016) and the European Union's Horizon 2020 research and innovation program under the Marie Skłodowska-Curie grant agreement no. 730888 (RESET). This work contributes to the Strategic Research Plan of the Centre for Marine Technology and Ocean Engineering (CENTEC), which is financed by the Portuguese Foundation for Science and Technology (Fundação para a Ciência e Tecnologia—FCT) under contract UIDB/UIDP/00134/2020.

**Data Availability Statement:** Not applicable.

**Conflicts of Interest:** The authors declare no conflict of interest.

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
