# Peer review of "Data-Driven Damage Quantification of Floating Offshore Wind Turbine Platforms Based on Multi-Scale Encoder–Decoder with Self-Attention Mechanism"

_jmse, doi:10.3390/jmse10121830_

Round 1
Reviewer 1 Report
The work concerns data-driven investigation on damage quantification of floating wind turbines. A learning approach based on a neural network is developed and used for case studies. Together with the Square Euclidean distance, the method exhibits a superior performance than other industry-adopted variants. Overall, the paper is written in a clear manner. The literature review is comprehensive. The numerical experiments are properly designed and presented. I think the work is a good addition to the existing literature regarding the damage issue on floating wind turbines. To this end, I recommend it under the consideration of publication.
Nevertheless, there are a few minor issues as follows.
1. In section 2.1 around line 180, the maths forms of the powers are incorrect. 2. The figures and the tables seem to be not centered.
Author Response
Dear Editor and Reviewers:
We would like to seize this opportunity to thank the Editor and Reviewers for their comments and suggestions on our manuscript entitled “Data-driven Damage Quantification of Floating Off-shore Wind Turbine Platforms based on Multi-Scale Encoder-Decoder with Self-Attention Mechanism”. We find these comments and suggestions very helpful and valuable in enhancing the quality of our paper. We have made appropriate changes in response to the reviewers’ comments and suggestions. The revised version of the manuscript, with changes marked in red, is enclosed. Detail of response to the comments is presented below:
Response to the reviewer's comments:
Reviewer #1:
Comment 1. The work concerns data-driven investigation on damage quantification of floating wind turbines. A learning approach based on a neural network is developed and used for case studies. Together with the Square Euclidean distance, the method exhibits a superior performance than other industry-adopted variants. Overall, the paper is written in a clear manner. The literature review is comprehensive. The numerical experiments are properly designed and presented. I think the work is a good addition to the existing literature regarding the damage issue on floating wind turbine. To this end, I recommend it under the consideration of publication.
Nevertheless, there are a few minor issues as follows. 1. In section 2.1 around line 180, the math forms of the powers are incorrect. 2. The figures and the tables seem to be not centered.
Response: The mistakes around line 180 and in some figures and tables have been corrected in the revised version.

Reviewer 2 Report
The manuscript equipped with similarity measurement to enhance its capability to accurately quantify damage effects from different scales of coded features using raw platform responses and without human intervention. The comaprisons highlight the superiority of MSCSA-AED in complex operating conditions. Some problems should be addressed befor accepting:
(1) The motivations of the research is confuse. A comprehensive overview should be provided to highlight the research motivations.
(2) Compared with the extant models, the superiority of the proposed model should be discussed.
(3) In my opinion, the comaprison analysis is simple, authors should enhance it. Meanwhile, the complxity of the proposed model should be explored.
(4) In conclusion, the limitations and further prespects should be discussed deeply,
(5) The references should be revised carefully. Such as,[30] and [31] occure twice. some references without Page information should provide the DOI.
(6) The linguistic quality of the manuscript should be polished.
Author Response
Dear Editor and Reviewer:
We would like to seize this opportunity to thank the Editor and Reviewer for their comments and suggestions on our manuscript entitled “Data-driven Damage Quantification of Floating Off-shore Wind Turbine Platforms based on Multi-Scale Encoder-Decoder with Self-Attention Mechanism”. We find these comments and suggestions very helpful and valuable in enhancing the quality of our paper. We have made appropriate changes in response to the reviewers’ comments and suggestions. The revised version of the manuscript, with changes marked in red, is enclosed. Detail of response to the comments is presented below:
Response to the reviewer's comments:
Reviewer #2:
The manuscript equipped with similarity measurement to enhance its capability to accurately quantify damage effects from the different scales of coded features using raw platform responses and without human intervention. The comparisons highlight the superiority of MSCSA-AED in complex operating conditions. Some problems should be addressed before accepting:
Comment 1. The motivation of the research is confused. A comprehensive overview should be provided to highlight the research motivation.
Response: We have clarified our motivation in the revised manuscript. Our motivation is to develop a framework data-driven damage quantification of offshore wind turbine structure based on unsupervised learning to reduce dependence on human intervention in damage feature extraction. This is essential because floating offshore wind turbines operate in remote locations, hence reducing human intervention would reduce the levelised cost of electricity and maintenance costs. In addition, existing data-driven (quantitative) damage models are predominantly built on supervised learning, which may not be particularly suitable for floating offshore wind turbine damage quantifications for maintenance. In our manuscript, it is noted that both the application of unsupervised learning in damage quantification and its robustness in floating wind turbines are yet to be investigated. Therefore, our study provides a timely contribution to knowledge is this area and sets out to achieve damage quantification through unsupervised learning to avoid priori dependencies introduced due to labels.
Comment 2. Compared with the extant models, the superiority of the proposed model should be discussed.
Response: Monotonicity is used to evaluate the performance of the quantified damage to show the superiority of the proposed method for damage quantification. Some new discussion is modified in the revised version about the comparison. We have added the following text and figure in the manuscript.
“Monotonicity is used as a metric to evaluate the performance of the quantified damage.
|
Figure 7: Monotonicity of the quantified damage using three models |
As shown in Figure 7, the damage quantified by the proposed MSCSA-AED model offers the highest monotonicity score. Generally, the higher the damage monotonicity indicator, the more reliable and easier for the prognosis to be conducted. Hence, the comparative results indicate that MSCSA-AED (MSCSA) has a monotonicity of 0.6349. This is higher than the monotonicity of both DCN and MSCNN by at least 22.49%, indicating that the proposed method is well-suited to produce reliable damage quantification.”
Comment 3. In my opinion, the comparison analysis is sample. Authors should enhance it. Meanwhile, the complexity of the proposed model should be explored.
Response: We have added more analyses in the revised manuscript, including on complexity and monotonicity of the models.
“Analysis of the models’ complexity is presented in Table 2.
Table 2 Complexity analysis of the models |
|
Method Name |
Complexity index |
DCN-AED |
O(4.7×105) |
MSCNN-AED |
O(3.3×107) |
MSCSA-AED |
O(1.7×107) |
The computational complexity of the proposed method is significantly higher than DCN, but lower than MSCN because of the introduction of recurrent neural networks in the feature resampling. The proposed method can obtain better performance at a relatively low complexity when it is combined with the performance analysis results of the quantified impairment to further demonstrate its superiority”
Comment 4. The references should be revised carefully. Such as 30 and 31 occur twice. Some references without page information should provide the DOI.
Response: We have reformatted all references using a standard Harvard-modified citation style.
Comment 5. In conclusion, the limitations and further presents should be discussed deeply.
Response: We have added limitations and future work in the conclusions, as advised.
Comment 6. The linguistics quality of the manuscript should be polished.
Response: We have revised the manuscript and improved the linguistic quality and writing style.

Reviewer 3 Report
The authors studied the quantification of damage on a multibody floating offshore wind turbine employing a novel framework of MSCSA-AED. Only raw response data and no human intervention are involved in the damage estimation, which has a good application prospect. The paper is innovative and well-written. But there are some small problems to be further confirmed before publication.
(1) In general, the learning rate should increase with the larger mini-batch size to ensure the convergence of the model. But the learning rates in Table 1 are fixed for four cases. Whether the hyperparameters are sufficiently optimized?
(2) The MSCSA-AED is trained by data-driven in this paper. If any physical mechanisms of platform responses could be considered in the training process of MSCSA-AED in the future works?
(3) In my opinion, the titles of Subsection 4.2 and 4.3 should be more specified to distinguish them better.
Author Response
Dear Editor and Reviewer:
We would like to seize this opportunity to thank the Editor and Reviewer for their comments and suggestions on our manuscript entitled “Data-driven Damage Quantification of Floating Off-shore Wind Turbine Platforms based on Multi-Scale Encoder-Decoder with Self-Attention Mechanism”. We find these comments and suggestions very helpful and valuable in enhancing the quality of our paper. We have made appropriate changes in response to the reviewers’ comments and suggestions. The revised version of the manuscript, with changes marked in red, is enclosed. Detail of response to the comments is presented below:
Response to the reviewer's comments:
Reviewer #3:
The author studied the quantification of damage on multibody floating offshore wind turbine employing a novel framework of MSCSA-AED. Only raw response data and no human intervention are involved in the damage estimation, which has a good application prospect. The paper is innovative and well-written. But there are some small problems to be further confirmed before publication.
Comment 1. In general, the learning rate should increase with the larger mini-batch size to ensure the convergence of the model. But the learning rates in Table 1 are fixed for four cases. Whether the hyperparameters are sufficiently optimized?
Response: In this study, the hyperparameters are sufficiently optimized by using more cases for the examination, especially for the model parameters.
Comment 2. The MSCSA-AED is trained by data-driven in this paper. If any physical mechanisms of platform responses could be considered in the training process of MSCSA-AED in the future works?
Response: We certainly will consider this advice in our future works. We plan to investigate the influence on embedding physics model into a data-driven model to improve its performance and robustness. We have added this in recommendation for future work in the conclusions.
Comment 3. In my opinion, the titles of subsection 4.2 and 4.3 should be specified to distinguish them better.
Response: We have modified Sections 4.2 and 4.3 as advised.

Reviewer 4 Report
Authors are suggested to include comparative analysis with benchmark deep learning algorithms like LSTM and CNN.
Author Response
Dear Editor and Reviewer:
We would like to seize this opportunity to thank the Editor and Reviewer for their comments and suggestions on our manuscript entitled “Data-driven Damage Quantification of Floating Off-shore Wind Turbine Platforms based on Multi-Scale Encoder-Decoder with Self-Attention Mechanism”. We find these comments and suggestions very helpful and valuable in enhancing the quality of our paper. We have made appropriate changes in response to the reviewers’ comments and suggestions. The revised version of the manuscript, with changes marked in red, is enclosed. Detail of response to the comments is presented below:
Response to the reviewer's comments:
Reviewer #4:
Comment 1: Authors are suggested to include comparative analysis with benchmark deep learning algorithms like LSTM and CNN.
Response: We have already compared our model with two CNN-based state-of-art auto-encoder-decoders in section 4.2 of the original manuscript. In the MSCNN based AED model, the connection between the down-sampling and up-sampling features is implemented via the LSTM network.
On the need to conduct further analysis, we sincerely appreciate the reviewer’s advice. However, developing an AED based on LSTM is outside the scope of this manuscript and we intend to explore as part of our further studies.

Round 2
Reviewer 2 Report
The manuscript has revised very well according to reviewers' comments. In my opinion, I recommend the manuscript to be published in JMSE.
Reviewer 4 Report
Okay. Good work. Please improve language wherever required.